# Assessing the potentials of bacterial antagonists for plant growth promotion, nutrient acquisition, and biological control of Southern blight disease in tomato

Farjana Sultana[1], M. Motaher Hossain[2]*

**1** College of Agricultural Sciences, International University of Business Agriculture and Technology, Dhaka, Bangladesh, **2** Department of Plant Pathology, Bangabandhu Sheikh Mujibur Rahman Agricultural University, Gazipur, Bangladesh

* hossainmm@bsmrau.edu.bd

**Data Availability Statement:** All relevant data are within the paper and its Supporting Information files.

## Abstract

Southern blight of tomato caused by *Sclerotium rolfsii* can cause severe plant mortality and yield losses. The use of rhizobacteria for the biological control of Southern blight disease is a potent alternative to chemical fungicides. Although rhizobacteria are prolific candidates, comprehensive reports regarding their use in tomato disease management are limited. The present study screened six rhizobacterial strains for antagonism against *S. rolfsii* in dual culture and culture filtrate assays. The selected promising strains were tested further for plant-growth-promoting and biocontrol potentials under *in vitro*, greenhouse, and field conditions. Of the six strains screened, *Stenotrophomonas maltophilia* PPB3 and *Bacillus subtilis* PPB9 showed the superior performance displaying the highest antagonism against *S. rolfsii* in dual culture (PPB3 88% and PPB9 71% inhibition), and culture filtrate assays (PPB3 53–100% and PPB9 54–100% inhibition at various concentrations). Oxalic acid produced by *S. rolfsii* was significantly inhibited by both rhizobacteria and supported their growth as a carbon source. The strains produced hydrogen cyanide, chitinases, siderophores, biofilm, and indole acetic acid. They showed the potential to solubilize phosphate and fix nitrogen. Seed treatment with *S. maltophilia* PPB3 and *B. subtilis* PPB9 improved seed germination and tomato seedling vigour. Significant increases in plant growth, chlorophyll contents, and N, P, and K concentrations were attained in bacterized plants compared to non-treated controls. The application of antagonists on container-grown seedlings in a greenhouse environment and field-grown tomato plants reduced symptoms of damping-off and Southern blight. The sclerotial counts decreased significantly in these soils. Bacteria-inoculated plants had a higher yield than those in the non-treated control. Bacteria colonized the entire roots, and their populations increased significantly in the protected plants. The results show the potential capabilities of *S. maltophilia* PPB3 and *B. subtilis* PPB9 for growth promotion, nutrient acquisition, and biocontrol of southern blight disease in tomatoes.

**Funding:** Ministry of Science and Technology, Bangladesh.

**Competing interests:** The authors have declared that no competing interests exist.

## Introduction

Tomato (*Solanum lycopersicum* L.) is affiliated to the Solanaceae family that includes a large number of economically essential vegetables. It was native to South America, and domestication has occurred in Central America. Europeans distributed the tomato from the Americas to other parts of the world during the 16th century. Presently, this plant is extensively grown worldwide and has become a significant vegetable crop. Based on the worldwide production, tomato currently ranks 7th after maize, rice, wheat, potatoes, soybeans, and cassava [1]. The increasing trend in tomato production and consumption is continuing. This is because tomato fruits can be consumed as raw, cooked, processed, and have good palatability. Tomato fruits have a significant role in human nourishment. It is an abundant source of vitamins, minerals, fibers, and various phytochemicals that provide vital human health and immunity benefits against various disorders. Lycopene, a low-calorie antioxidant element, is the primary carotenoid found in tomatoes [2]. Studies show that lycopene helps reduce prostate cancer risk in test animals [3]. Thus, tomato consumption may be considered beneficial for the prevention of the development or progression of cancer. Scientific evidence is also available to correlate tomato intake with low cardiovascular disease risks [4].

The global demand for tomatoes is increasing due to rapid population growth. Achieving higher agricultural production in current farming practices is a big challenge. Moreover, tomato plants are vulnerable to an extensive range of abiotic and biotic aggravations. Plant pathogens constitute one of the vital biotic intimidations to tomato cultivation affecting both yield and quality. Worldwide, losses due to plant diseases (excluding viruses) are appraised to be approximately 40% of achievable tomato yield without crop protection [5]. Tomato is extremely susceptible to fungal pathogens that are causing early blight, late blights, Fusarium wilt, and southern blight. Southern blight caused by *Sclerotium rolfsii* is a highly devastating disease commonly found in subtropics and tropics. *Sclerotium rolfsii* is a soil-inhabiting omnivorous fungal pathogen infecting a wide range of vegetables, including tomato. The fungus persists for many years as sclerotia in the soil or on diseased crop debris. The soil-borne inoculum of the fungus affects seed germination and causes damping off. The fungus also infects the stem at or near the soil line causing necrotic rot. The rapidly developing lesions girdle the stem, leading to wilting of the plant abruptly and permanently [6]. Whitish mycelium and abundant sclerotia develop on rotting tissues. The pathogen can also cause rots of fruit in contact with the soil [9]. The disease often results in crop losses when soil and weather factors are favourable for disease development. Environmental conditions that favour *Sclerotium* disease development are high temperatures (27 to 35°C), humid conditions, and acidic soil [7]. Moreover, production of oxalic acid by *S. rolfsii* is considered an essential component of its pathogenesis factors. The fungal oxalic acid promotes polygalacturonase activity and creates an acidic environment in plant tissues. These, in turn, inactivate the prohibitins and phytoalexins and reduce host resistance to the pathogen [8].

Applying synthetic fertilizers and fungicides has been perceived as the core management strategy for maintaining proper growth and health in tomato plants. However, the current intensive agricultural methods often warrant an excessive application of these chemicals in the crop field and amplify the environmental burden, leading to irredeemable risks for the ecosystems and human health. Non-specific chemicals affect the beneficial microbial species in the soil and cause severe intimidation to soil fertility. Persistent chemicals are particularly hazardous and become rapidly concentrated in the food chain. Ingestion or exposure to these chemicals may result in acute or long-term health problems. Inconsistent and repeated application of the same active compound may also lead to developing a fungicide-resistant pathogen population, making it challenging to manage plant diseases effectively [9]. Hence, current chemical-

based agriculture has to be replaced with sustainable practices that carriage fewer public health risks and environmental issues. Most of all, the chemical control of *Sclerotium* diseases is exceptionally challenging due to its soil-borne nature. Counter to these complex and inefficacious chemical-based management strategies is the antipathogenic microorganisms. Besides providing various growth benefits to plants, the microbial antagonists are highly efficient in suppressing soil-borne diseases [10,11].

The rhizosphere is a dynamic habitat for large groups of beneficial microorganisms. It is well known that the rhizosphere microorganisms, directly and indirectly, benefit plant growth and development. The rhizobacteria that actively colonize plant roots and enhance plant growth and yield are referred to as plant growth-promoting rhizobacteria (PGPR) [12]. PGPR are widely used to improve plant growth and suppress plant diseases biologically. Thus, PGPR application in the tomato field for controlling *S. rolfsii* is considered an eco-friendly practice that ensures the higher crop productivity and implements a sustainable approach that manages disease without risk. However, the PGPR that inhibit oxalic acid production of *S. rolfsii* are considered perfect candidates for incorporating them as biocontrol agents into the southern blight disease management program in tomatoes. Although numerous strains of PGPR have so far been reported with potential application in tomato cultivation, a few of them are described with oxalic acid-inhibiting abilities. In this view, the present study screened six PGPR strains for their efficacy against *S. rolfsii* to select the superior antagonists. Further studies were undertaken to evaluate the beneficial effects of the promising rhizobacteria on plant growth promotion, nutrient acquisition, and biological control of Southern blight disease in tomatoes. The ultimate goal is to develop PGPR-based bioagents that effectively reduce southern blight disease and enhance tomato yields.

## Materials and methods

### Bacterial collection source

Six bacterial strains *Pseudomonas stutzeri* PPB1, *Bacillus subtilis* PPB2, *Stenotrophomonas maltophilia* PPB3, *B. amyloliquefaciens* PPB4, *B. subtilis* PPB5, and *B. subtilis* PPB9 previously isolated from the cucumber rhizosphere (*Cucumis sativus* L. cv. Baromashi) [12] were used as bio-inoculants in this study. Nutrient broth (NB) was amended with 15% glycerol to maintain the bacterial stock cultures at -20˚C. During bioassays, active cultures were obtained by streaking the stock cultures onto the nutrient agar (NA) plates, followed by incubating at 28˚C for 48 hours.

### Phytopathogen cultures

The plant pathogens *Rhizoctonia solani* AR-01, *Phytophthora capsici* PPC-1, *Sclerotinia sclerotiorum* PSB-1, and *Sclerotium rolfsii* SR-1 were used in this study. These fungi were taken from the Department of Plant Pathology Stock Cultures, Bangabandhu Sheikh Mujibur Rahman Agricultural University (BSMRAU), Gazipur, Bangladesh. The fungus cultures were stored in potato dextrose agar (PDA) slant at 4˚C until use. In order to use in the bioassays, pieces of stock cultures were placed onto PDA plates and incubated at 25±2˚C for five days.

### Screening of bacteria for antagonistic activity against *Sclerotium rolfsii* SR-1

**Dual culture assay.** The antifungal activity of six bacterial isolates, *P. stutzeri* PPB1, *B. subtilis* PPB2, *S. maltophilia* PPB3, *B. amyloliquefaciens* PPB4, *B. subtilis* PPB5, and *B. subtilis* PPB9, against *S. rolfsii* SR-1 was determined by the dual culture technique [13]. Bacterial

cultures were streaked onto glucose casamino yeast extract medium (GCY) (1.5% glucose, 0.15% casamino acid, 0.1% yeast extract, 0.15% $K_2HPO_4$, 0.1% $MgSO_4.7H_2O$, 2.0% agar, pH 7.2) and incubated for 24 h. GCY plates amended with Provax-200 (Carboxin, 5,6-dihydro-2-methyl-1,4-oxathin-3-carboxanilide) (Hossain Enterprise CC Limited, Dhaka, Bangladesh) at a concentration of 200 ppm were included as a positive check. Provax-200 was added in an autoclaved GCY medium following the poisoned-food technique before the fungal inoculation. The GCY plates without bacterial inoculation or fungicidal amendment were treated as negative controls. Later, a mycelial disk (5-mm in diameter) excised from a new actively growing colony of *S. rolfsii* SR-1 was placed onto the center of the Petri plates. All treatments were prepared in triplicates. The inoculated plates were incubated at 28°C in an incubator. After five days, when *S. rolfsii* SR-1 in control plates covered the entire plate, the fungal radial growth diameter in dual cultures was recorded. The percent growth inhibition (PI) of *S. rolfsii* SR-1 due to the bacterial antagonistic activity over the control was calculated using the following formula:

$$\% \text{ Inhibition of growth} = \frac{X - Y}{X} \times 100$$

Where,
X = Mycelial growth of the pathogen in the absence of bacteria.
Y = Mycelial growth of the pathogen in the presence of bacteria.

The percent of mycelial growth inhibition of *S. rolfsii* by the fungicide was calculated using the same formula, where the mycelial growth of the pathogen in the absence (X) and the presence of Provax-200 (Y) was measured.

**Culture filtrate assay.** The bacterial isolates listed in this study were further evaluated for antifungal activity of their culture filtrates against *S. rolfsii* isolate SR-1. Bacteria were cultured for five days in Yeast Peptone (YP) broth on a shaker at 120 rpm at 28°C. The broth was centrifuged at 15000 rpm at 4°C, and supernatants were collected. The collected supernatants were passed through 0.45 μm membrane filters. The cell-free filtrates were added to the autoclaved GCY medium at the rate of 0%, 10%, 25%, or 50% (v/v). GCY plates amended with Provax-200 (Carboxin, 5,6-dihydro-2-methyl-1,4-oxathin-3-carboxanilide) (Hossain Enterprise CC Limited, Dhaka, Bangladesh) at a concentration of 200 ppm were included as a positive check. Mycelial plugs of *S. rolfsii* isolate SR-1 from actively growing margin were transferred onto Petri dishes and incubated at 28°C for five days in an incubator. All treatments were prepared in triplicates. The percentage inhibition of the test pathogen was calculated as described previously.

**Inhibition of oxalic acid production assay.** The efficacy of antagonists was tested against oxalic production by the fungus. An amount of 100 ml potato dextrose broth (PDB) was prepared in 250 ml Erlenmeyer flasks and autoclaved. Each flask was co-inoculated with ten mycelial disks (5-mm in diameter) excised from the fresh colony of *S. rolfsii* SR-1a 9 mm and a loopful of the overnight culture of each antagonist. PDB inoculated with *S. rolfsii* alone served as a control. Three replications were taken for each culture and incubated in an incubator at 28°C for two weeks. The broth containing mycelial growth was filtered through Whatman filter paper No.1 to remove the mycelial mass, which was later dried and weighed. The culture filtrates were centrifuged at 4500 rpm for 20 min and filtered by a 0.45 μm Millipore filter to obtain cell-free filtrates. Five milliliters of each cell-free filtrate were taken in a 15 ml sterilized centrifuge tube, and 4 ml of calcium chloride acetate buffer was added [14]. The mixtures were centrifuged at 4,500 rpm for 10 min, and the supernatants were thrown away. The deposits were washed with 5 ml of 5% acetic acid saturated with calcium oxalate and recentrifuged. Each deposit was dissolved in 5 ml of 4 N $H_2SO_4$ in a 100 ml conical flask and heated in a

water bath at 80–90˚C. Finally, the hot solutions were filtrated with 0.02 N potassium permanganate until faint pink color formed. One ml of 0.02 N $KMNO_4$ reacted with 1.2653 mg of oxalic acid. The oxalic acid production was calculated [14] and expressed as mg/g of dry fungal biomass.

## Characterization of elite bacteria for oxalotrophic traits

The ability of elite antagonists to utilize oxalic acid as carbon was tested. Active cultures of the bacteria were obtained by streaking the stock cultures onto the NA plates, followed by incubating at 28˚C overnight. Borosilicate glass test tubes containing 20 ml water-yeast broth (WYB, 5 g $l^{-1}$ NaCl, 0.05 g $l^{-1}$ yeast extract, 1 g $l^{-1}$ $KH_2PO_4$) supplemented with either oxalic acid or citric acid as a carbon source at concentrations of 0.5 mM, and 5.0 mM were prepared in triplicate for each culture. Citric acid was used as a positive check. Media prepared without organic acid addition were included as controls for growth supported by yeast extract. After autoclaving, the glass tubes containing broths were inoculated with a single bacterial colony and incubated on a shaker at 120 rpm at 28˚C for four days. The optical density of the cultures was measured daily at 0, 1,2, 3-, and 4-days post-inoculation.

## Characterization of elite bacterial isolates for plant-growth-promoting and biocontrol traits

**Phosphate solubilization.** The phosphate solubilization ability of the two bacterial isolates was evaluated using Pikovskaya agar plates [15]. Briefly, bacteria were cultured in NB on a shaker at 120 rpm at 28˚C for 48 hours. An aliquot of 10 µl of the bacterial culture was spot inoculated on Pikovskaya agar plates and subsequently incubated in an incubator at 37˚C. After six days, the plates were visually checked for a clear zone around the bacterial colonies.

**Indole-3-acetic acid (IAA) production.** The bacterial isolates were cultured in NB amended with 0.1% tryptophan in an incubator at 37˚C for 72 hours. The culture was centrifuged at 15,000×g for 10 min, and the supernatant was collected. A suspension was prepared by mixing 1 ml of the supernatant with two drops of ortho-phosphoric acid and 4 ml of Salkowski reagent (mixer of 50 ml of 35% of perchloric acid and 1 ml of 0.5 M $FeCl_3$) and incubated in the dark for 20 min. The absorbance of the solution was measured at 530 nm and plotted on the IAA standard curve.

**Nitrogen fixation ability.** Bacterial nitrogen fixation ability was assayed following the method described by Ker [16] with slight modifications. Each rhizobacterial strain was cultured in Yeast Extract Peptone broth on a shaker at 120 rpm at 28˚C for 24 hours. The culture was diluted to $10^7$ CFU $ml^{-1}$, and an aliquot of 2 µl of the diluted culture was taken for spot inoculation (4 spots $plate^{-1}$) onto N-free solid LG medium [19]. The inoculated plates were incubated at 28˚C under static conditions for ten days. The non-inoculated plates incubated at the same condition were used as controls. The growth of colonies in the N-free LG medium designated a positive nitrogen fixation ability of the bacterium.

**Siderophore production.** Chrome Azurol S (CAS) assay was used to determine the siderophore production ability of the bacteria [17]. At first, CAS (0.06 g) was dissolved in deionized water (50 ml) and slowly mixed with iron III solution (0.0027 g of $FeCl_3$-$6H_2O$ in 10 ml of 10 mM HCl) and hexadecyl-trimethyl-ammonium bromide (HDTMA) (0.73 g dissolved in 40 ml water). Piperazine-N,N'-bis(2-ethanesulfonic acid) (PIPES) (32.24 g) was dissolved in 900 ml water, in which a solution of 50% diluted NaOH (12 g) was added to raise the pH to 6.8. After autoclaving, both were mixed, and the resultant CAS broth was stored until use. Each bacterium was grown overnight in YP broth on a shaker at 120 rpm at 28˚C. The culture was

diluted to $10^7$ CFU ml$^{-1}$. Luria Broth (LB) agar plates (10 cm) were prepared, and each plate was divided into four equal sectors. Then 10 µl culture of each rhizobacterium strain was spotted in the center of each sector in the LB agar plates and incubated at 28˚C for one day. Later, 15 ml CAS broth was overlayed over those LB agar plates. The plates were incubated in an incubator at 37˚C for 12 hours. The formation of orange halo areas around the bacterial colonies on the blue background indicated a positive reaction for siderophore production by the bacteria.

**Chitinase production.** Bacteria were cultured in NB overnight on a shaker at 120 rpm at 28˚C. Chitin plates were prepared by amending M9 agar medium with 1% (w/v) colloidal chitin. The plates were divided into four equal sectors, and each sector was spot (4 spots plate$^{-1}$) inoculated with 10 µl of overnight grown culture. The inoculated plates were incubated at 37˚C for 96 hours. The formation of the clear zone around bacterial colonies implied a positive reaction for chitinase production [18].

**Hydrogen cyanide (HCN) production.** Bacteria were cultured in NB overnight on a shaker at 120 rpm at 28˚C. Nutrient agar was prepared by supplementing with 0.44% (w/v) glycine and plated in 9 cm Petri dishes. The agar surface was streak-inoculated with the bacterial culture. A sterile Whatman filter paper No. 1 drenched in filter sterile 2% (w/v) sodium carbonate in 0.5% (v/v) picric acid was placed on top of the culture. The plates were incubated in an incubator at 30˚C for 72 hours. The change in color of the overlaid filter paper from yellow to orange, red, or brown implied lesser, moderate, or higher HCN production levels, respectively.

**ACC deaminase assay.** Dworkin and Foster (DF) minimal salts supplemented with 1-aminocyclopropane-1-carboxylic acid (ACC) as the only nitrogen source were prepared. The bacteria were cultured overnight in NB medium and collected by centrifugation. The bacteria were then inoculated into the DF-ACC medium and incubated at 30˚C in a shaker at 160 rpm for 48 hours. The uninoculated DF-ACC medium was used as the control. The culture was centrifuged at 15,000×g at 4˚C, and then the supernatant was diluted in test tubes with a DF medium at a 100:1 ratio. Two milliliters of ninhydrin reagent were added to each tube. The tubes were shaken and placed in a hot water bath for 30 min. The solution then turned purple. The boiled sample was left at 30˚C for another 10 min before measuring the absorbance at 570 nm. In this assay, the DF-ACC medium was used as a blank.

***In vitro* antagonistic activity against a broad spectrum of phytopathogens.** The dual culture technique was employed to assess the *in vitro* antagonism of the selected bacteria against several phytopathogens, *Rhizoctonia solani* AR-01, *Phytophthora capsici* PPC-1, and *Sclerotinia sclerotiorum* PSB-1. The mycelial plugs of 5-mm diameter were obtained from the 5-day old culture of these pathogens and individually placed onto the center of the PDA plate. An overnight bacterial culture was then streak inoculated at an equidistance of 3 cm from the fungal plug. The dual inoculated plates were incubated at 28˚C for seven days. The control cultures of phytopathogens were grown without bacteria. Inhibition of the fungal mycelial growth in dual culture with bacteria relative to the control indicating the antagonistic activity of the bacteria was measured.

***In vitro* biofilm formation.** The bacterial ability to produce biofilm was tested during *in vitro* growth of the bacteria. The isolates were grown in NB at 28˚C on a shaker at 180 rpm overnight. The cultures were diluted to a ratio of 1:100. Fifty microliters of the diluted culture were inoculated into borosilicate glass test tubes containing 5 mL of SOBG [19]. The inoculated glass tubes were incubated at 28˚C in static conditions for 72 hours. The pellicle attached to the broth surface was gently harvested from the glass tubes, washed thrice with sterile distilled water, and quantified at 600 nm [19].

## Inoculum preparation of bacteria for tomato seed and root treatment

Bacteria were cultured in 250 ml Erlenmeyer flasks containing 100 ml Yeast Peptone (YP) broth on a shaker at 120 rpm overnight at 28˚C. The broth was centrifuged at 15000 rpm at 4˚C, and bacterial pellets were collected. Each pellet was washed thrice with sterile distilled water. The purified pellet was suspended in 1.5 ml sterile distilled water by vortexing for a few seconds. Approximately 0.65 g tomato seeds (200 seeds) were surface sterilized with 5% (v/v) sodium hypochlorite solution (NaClO) for 3 min, followed by several washes in sterile water and air-dried. Dry seeds were soaked in the bacterial suspension ($OD_{600}$ = 1.0) and coated with bacterial cells by frequent stirring for 5 min. Seeds treated with sterile distilled water served as the control. The bacteria-coated seeds were scattered on a Petri dish and air-dried overnight at 25±2˚C. The number of bacteria on the seed surfaces counted by serial dilution technique before sowing was approximately $10^8$ CFU/seed. The bacterial suspension for root treatment was prepared by adjusting the final concentration of the bacteria to around $10^8$ CFU ml$^{-1}$ ($OD_{600}$ = 0.12).

## *In vitro* effect of bacteria on seed germination and vigour index in tomato

In order to determine the effect of selected bacteria on germination and seedling vigour, 200 bacteria-coated seeds of tomato cv. Mintoo Super F1 Year-round (Lal Teer Seed Company, Gazipur, Bangladesh) were tested. As a control treatment, an equal number of seeds treated with sterile water was prepared. Seeds were placed on two moistened filter paper layers in 9-cm Petri dishes, allotting 25 seeds in each plate. The Petri dishes were incubated at 28±2˚C in a light incubator. Sterile water was added to the Petri dishes every other day. The germination percentage and seedling length were recorded after seven days. The seedling length was measured from the root tip to the shoot tip on the fifteenth day of culture. Seedling vigour index was calculated from the following formula:

$$\text{Vigour index} = \% \text{ germination} \times \text{total plant length}$$

## *In vivo* effect of bacteria on the germination and growth of tomato

The selected rhizobacterial isolates (*S. maltophilia* PPB3 and *B. subtilis* PPB9) were tested for their ability to promote growth in tomato plants. Tomato seeds were treated with bacteria and water (control) as described above. The soil collected from the research field of the Department of Plant Pathology, BSMRAU, Gazipur, Bangladesh, was used as a potting medium. The texture of the soil was sandy loam, and the pH was 6.38 with 1.08% organic carbon (OC), 1.87% organic matter (OM), 0.27% nitrogen (N), 0.09% phosphorus (P), and 0.87% potassium (K). The soil was autoclaved twice at 24 h intervals at 121˚C and 15 psi for 20 minutes. For each treatment, 15 pots (11.50 cm × 15.0 cm) were taken. Each pot was filled with about 500 g of the sterilized soils and sown with five seeds. Pots were placed in a growth room at 24˚C temperature and a 16/8 h photoperiod for 42 days. The germination percentage in each treatment was recorded daily for 12 days. On day 13, seedlings were thinned to one per pot. Plants were watered on alternate days. Before harvest, three plants were randomly selected for each treatment to determine specific leaf area (SLA) and photosynthetic pigments. Three mature leaves per plant were scanned and weighed to determine the SLA = area (m$^2$)/dry weight (kg). Leaf area was calculated with the software ImageJ (v1.47). To determine leaf chlorophyll content, fresh leaf samples were collected from the same plants and washed with distilled water. Leaves were blotter dried and cut into about ~1 cm piece. One-gram leaf pieces were placed in 5 ml of 80% acetone (v/v) and incubated overnight in the dark. On the next day, the mix was agitated on a shaker until the leaves were completely bleached. The supernatant was collected by

centrifuging the solution for 10 minutes at 5000 rpm, which was used to measure total chlorophyll (a+b) at 663 and 645 nm absorbance. Pigment concentrations were calculated using the formula [20] given below:

$$\text{Total Chlorophyll } (a + b) \ (\text{mg/g FW}) = [20.2 \times (A645) - 8.02 \times (A663)] \times V/(1000 \times FW)$$

Where,

V = Final volume of 80% acetone (ml)

FW = Fresh weight of sample taken (g)

At harvest, the whole plant was gently uprooted, and the roots were washed in running tap water to remove the attached soil. Data were recorded on morphological parameters such as root and shoot length, the number of leaves, stem diameter, and shoot and root fresh and dry weights. For dry weight, shoots and roots were oven-dried at 65°C for four days and then weighed.

## Determination of nutrient elements

The oven-dried shoots were separated, ground, and prepared in triplicate. After nitric-perchloric acid (7:3) digestion of the shoot samples, the concentration of cations ($Na^+$, $K^+$, $Ca^{2+}$, and $Mg^{2+}$), iron, and total phosphorus was determined using inductively coupled plasma (ICP) spectrometry (Optima 4300DV, Perkin-Elmer, UK). Nitrogen concentration in the tomato shoots was determined using the Kjeldahl method [21].

## Preparation of wheat grain inoculum of *Sclerotium rolfsii* SR-1

Inoculum of *S. rolfsii* SR-1 was prepared by taking 100 g moistened wheat grain in a 500 ml Erlenmeyer flask. After plugging the mouth with cotton, the conical flasks were autoclaved at 121°C and 15 PSI for 20 min and allowed to cool on a clean bench. The grains were inoculated with 12 to15 mycelial disks (5 mm diameter) obtained from the actively growing margin of 5-day-old PDA cultures of *S. rolfsii* SR-1. The inoculated flasks were plugged with sterile cotton and incubated at 25°C in the dark for 2 to 3 weeks. The flasks were shaken weekly to allow for consistent growth of *S. rolfsii*. The culture continued until the fungus thoroughly colonized the surface of wheat grains. Colonized grains were air-dried at laboratory temperature (25±2°C) and stored at 4°C until further use.

## Biocontrol efficacy of PGPR against *Sclerotium rolfsii*

**Seedling test.** Tomato seeds were surface sterilized by immersing in 5% (v/v) sodium hypochlorite for 3 min, followed by three rinses with sterile distilled water. Bacterized seeds were prepared as described above. The seed treatment with Provax-200 was done at a concentration of 0.3% (w/w). Seeds lacking treatment with the antagonists and fungicide were considered control (unprotected control). Plastic seed trays (20×10×5 cm) were used to raise seedlings, allotting three trays for each treatment. The soil from the research field described in the preceding section was used as a potting medium. After autoclaving (121°C and 15 psi for 20 min) twice at one-day intervals, the soil was thoroughly mixed with colonized wheat grain inoculum of *S. rolfsii* at a concentration of 2.0% (w/w). The soil-inoculum mix was placed in sterilized seed trays (1.0 kg/tray) and moistened to approximately 50% water-holding capacity. After three days, each tray was sown with 30 tomato seeds, 10 in each row. Seed trays were placed in a growth room at 24°C temperature and a 16/8 h photoperiod for seven weeks. Plants were watered on alternate days. Seedlings were observed weekly for damping-off symptoms and signs of *S. rolfsii* until seven weeks after sowing. The incidence of damping-off in seedlings

was expressed as a percentage of the total seedlings with damping-off calculated from the following formula:

$$\% \text{ Seedlings with damping off} = \frac{(\text{No. of the seedlings with damping off})}{(\text{Total no. of seedlings in the tray})} \times 100$$

Additionally, the number of sclerotia formed on soil surfaces was counted and expressed per cm$^2$ surface area.

**Potted plant tests.** In this experiment, treatments included were a control (unprotected), a fungicide protected check, and two PGPR treatments. Surface sterilized tomato seeds were sown in seed trays (20×10×5 cm), and seedlings were raised in the net house for two weeks. The soil from the research field described in the preceding section was used as a potting medium. Autoclaved field soil was mixed thoroughly with colonized wheat grain inoculum of *S. rolfsii* at a concentration of 2.0% (w/w) and loaded in the pots (11.50 cm × 15.0 cm) (500 g/pot). Fifteen pots were prepared for each treatment. Seedlings were uprooted from seed trays, and bacterial treatment of the seedlings was given by dipping roots in the suspension ($10^8$ CFU ml$^{-1}$) of *S. maltophilia* PPB3 and *B. subtilis* PPB9 for 30 min. Seedlings of unprotected control were immersed in the sterilized water. Soils drenched with Provax 200 (0.3% w/v) at transplanting and one-week post transplanting were considered fungicide-protected checks. Pots were placed in a growth room at 24˚C temperature and a 16/8 h photoperiod for 12 weeks. Plants were watered on alternate days. Two weeks after planting, seedlings were watered weekly with 40 mL of soluble fertilizer solutions (0.6 g l$^{-1}$ of NPK solution 20: 20: 20) until 8 weeks after planting. Disease developments on each plant were rated every week from 2 weeks after transplanting using the following scale: 0 = no symptoms, 1 = <25% of leaves with symptoms, 2 = 26 to 50% of leaves with symptoms, 3 = 51 to 75% of leaves with symptoms, 4 = 76 to 100% of leaves with symptoms and 5 = plant dead. The disease index was calculated using the following rating by the formula:

$$\text{Disease index} = \frac{(\text{Rating No.} \times \text{No. of the plants in the rating}) \times 100}{(\text{Total no. of plants} \times \text{highest rating})}$$

Additionally, the number of sclerotia formed on soil surfaces was counted and expressed per cm$^2$ surface area.

**Field grown plant tests.** The PGPR strains were further evaluated against *S. rolfsii* at the experimental field, Department of Plant Pathology, Bangabandhu Sheikh Mujibur Rahman Agricultural University, Gazipur, Bangladesh. The soil characteristics were described in the preceding section (Sandy loam texture, pH 6.38, 1.08% OC, 1.87% OM, 0.27% N, 0.09% P, and 0.87% K). This experiment included an uninoculated negative control, a pathogen inoculated control, a fungicide protected chemical check, and two PGPR treatments. Experiments were laid in a randomized block design (RBD) with three replications for each treatment. The unit plot size was 2.0 m × 2.0 m, and the spacing between plots was 0.5 m. A spacing of 50 cm between rows and plants and a total of 12 plants per plot were assigned. The soil of the field was pulverized well by deep plowing. A recommended dose of 120 kg, 40 kg P,100 kg K, and 2 tons cow dung per hectare was applied. The full dose of P, K, and cow dung and one-third of N was applied at the time of final land preparation, and the remaining N was applied in two equal installments at 3 and 5 weeks after the transplanting of tomato. The colonized wheat grain inoculum of *S. rolfsii* was mixed with soil in inoculated micro plots (50 g inoculum/m$^2$ area). Plots amended with an equal amount of autoclaved wheat grain were considered as uninoculated negative controls. The soil drenched with Provax 200 (0.3% w/v) at transplanting and one-week post transplanting was treated as fungicide-protected check plots. Bacterial

treatment of the seedlings was given by dipping roots in the suspension ($10^8$ CFU ml$^{-1}$) of *S. maltophilia* PPB3 and *B. subtilis* PPB9 for 30 min before transplanting in the field. The seedlings dipped in the sterilized water for 30 min were considered as unprotected controls. After transplanting, weeding, watering, and other intercultural operations were done regularly. Disease developments on each plant were rated at 12 weeks after transplanting using the following scale: 0 = no symptoms, 1 = <25% of leaves with symptoms, 2 = 26 to 50% of leaves with symptoms, 3 = 51 to 75% of leaves with symptoms, 4 = 76 to 100% of leaves with symptoms and 5 = plant dead. The disease index was calculated using the following rating by the formula:

$$\text{Disease index} = \frac{(\text{Rating No.} \times \text{No. of the plants in the rating}) \times 100}{(\text{Total no. of plants} \times \text{highest rating})}$$

At the end of the experiment, data on plant height were recorded. Fruits were harvested at several spells, and yield was calculated.

### Root colonization

Studies were conducted to study the root colonization ability of the rhizobacteria in tomatoes. Seeds were treated and sown in pots as described above. For each treatment, 21 plants were grown in 21 pots for 12 weeks. Roots were harvested at 1, 2, 4, 5, 6, and 7 weeks after sowing. Root systems were thoroughly washed with running tap water to remove adhering soil particles. Roots were cut into top, middle, and bottom regions. Root pieces belong to each segment were weighed and then surface sterilized in 5% (v/v) sodium hypochlorite solution (NaClO) for 1 min, followed by several washes in sterile water [22]. Root tissues of each segment were separately homogenized in sterilized distilled water. Appropriate dilutions of the homogenates were plated onto Yeast Peptone Agar (YPA) media. The plates were incubated overnight at 28˚C, and the number of colony-forming units (CFU) per gram of root tissue was determined.

### Statistical analysis

A completely randomized design (CRD) was followed for all experiments. The data presented are from representative experiments that were repeated 2 to 3 times. Statistix 10.0.0.9 (Analytical Software, FL, USA) was used to compare treatments via ANOVA (analysis of variance) using the least significant differences test (LSD) at a 5% ($P \leq 0.05$) probability level.

### Results

### *In vitro* antagonistic activities of the bacterial isolates in dual culture assays

In dual culture assays, all six rhizobacterial isolates screened inhibited the growth of *S. rolfsii*, ranging from 58.22% to 88.29% (Table 1). Of these, maximum inhibition was recorded by isolate *Stenotrophomonas maltophilia* PPB3 (88.29%), followed by isolate *Bacillus subtilis* PPB9 (71.84%). A distinct growth inhibition zone with the pathogen was observed. In plates treated with Provax 200, a complete inhibition (100%) of the pathogen growth was observed.

### *In vitro* antagonistic activities of the culture filtrates

The cell-free filtrates of all isolates showed effective antagonistic activities against the test pathogen, significantly inhibiting the mycelial growth of *S. rolfsii* at all three concentrations compared to control (Table 1). The inhibition of test pathogen at 10%, 25% and 50% concentrations ranged from 19.75% to 54.43%, 44.24% to 100.00% and 62.89% to 100.00%, respectively. Culture filtrates of strain PPB3 and PPB9 at 25% and 50% concentrations completely (100%) inhibited the mycelial growth of *S. rolfsii*, similar to Provax 200. Culture

**Table 1. Inhibition of mycelial growth and oxalic acid production of *Sclerotium rolfsii* by bacterial antagonists.**

| Antagonists | Inhibition of *S. rolfsii* in dual culture (%)* | Inhibition of S. rolfsii1 at different concentrations of culture filtrate of antagonists (%) | | | Oxalic acid production by *S. rolfsii* (mg/g fungal mass)** |
|---|---|---|---|---|---|
| | | 10% | 25% | 50% | |
| **Control** | 0.00±0.00a*** | 0.00±0.00a | 0.00±0.00a | 0.00±0.00a | 415.73±2.67d |
| ***Pseudomonas stutzeri* PPB1** | 58.22±0.96b | 19.75 ±0.13b | 44.24 ±0.14b | 62.89 ±0.65b | 403.86±4.76d |
| ***Bacillus subtilis* PPB2** | 61.84±0.83c | 45.78±0.35e | 67.38 ±0.00d | 100.00 ±0.00d | 183.71±3.29a |
| ***Stenotrophomonas maltophilia* PPB3** | 88.29 ± 0.51f | 53.48±2.18f | 100.00 ±0.00f | 100.00 ±0.00d | 176.30±3.25a |
| ***B. amyloliquefaciens* PPB4** | 62.82±0.77c | 34.29±0.51c | 54.34 ±0.45c | 68.57±0.71c | 346.78±5.27c |
| ***B. subtilis* PPB5** | 66.97±1.48d | 37.58 ±0.53d | 70.76 ±1.01e | 100.00 ±0.00d | 305.04±2.59b |
| ***B. subtilis* PPB9** | 71.84±0.95e | 54.43±0.60f | 100.00 ±0.00f | 100.00 ±0.00d | 175.49±4.02a |
| **Provax 200** | 100.00±0.00g | 100.00 ±0.00g | 100.00 ±0.00f | 100.00 ±0.00d | NT**** |

* The antagonistic activity of bacteria and their culture filtrates against *Sclerotium rolfsii* was measured as percent inhibition of radial growth of the fungal pathogens by antagonists in a dual plate assay. Values are means ± SE (n = 3).

** Oxalic acid production by *S. rolfsii* was quantified after culturing for two weeks. Values are means ± SE (n = 3).

***Different letters denote significant differences among treatments for each column according to Fisher's LSD test (*P* < 5%).

****NT = Not tested.

filtrates of strain PPB2 and PPB5 at 50% concentrations caused complete mycelial growth inhibition (100%) of *S. rolfsii.*

## Inhibition of oxalic acid production by bacteria

Five of the six rhizobacteria significantly reduced the oxalic acid production of *S. rolfsii* in dual broth cultures (Table 1). In the absence of rhizobacteria, *S. rolfsii* isolate SR-1 produced 415.73 mg oxalic acid/g of fungal biomass, while the quantity of oxalic acid in co-cultures with the antagonists ranged from 403.86 to 175.49 mg/g of fungal biomass. Of the five strains that reduced oxalic acid production, the lowest quantity of oxalic acid (175.49 mg/g) was recorded in dual culture with PPB9, which was statistically similar to those with PPB3 (176.30 mg/g) and PPB2 (183.71 mg/g). Since isolate PPB3 and PPB9 and their culture filtrates exhibited the highest inhibition of mycelial growth and maximum degradation of oxalic acid production by the test pathogen, these two isolates were selected as elite antagonists against *S. rolfsii* for further studies.

## Consumption of oxalic acid by bacteria

Compared to the control (without extra added carbon), we observed a small but significant increase in bacterial cell densities of *S. maltophilia* PPB3 and *B. subtilis* PPB9 in WYB with citric acid and oxalic acid as the added carbon source (5.0 mM and 0.5 mM) one day after inoculation (Fig 1). On day 2, day 3, and day 4, significant increases in cell densities of both bacteria were observed in treatments with citric acid and oxalic acid at a concentration of 5.0 mM compared to the control (S1 Table in S1 File). However, with the addition of citric acid and oxalic at a 0.5 mM concentration in WYB, we found little or no significant increase in cell densities

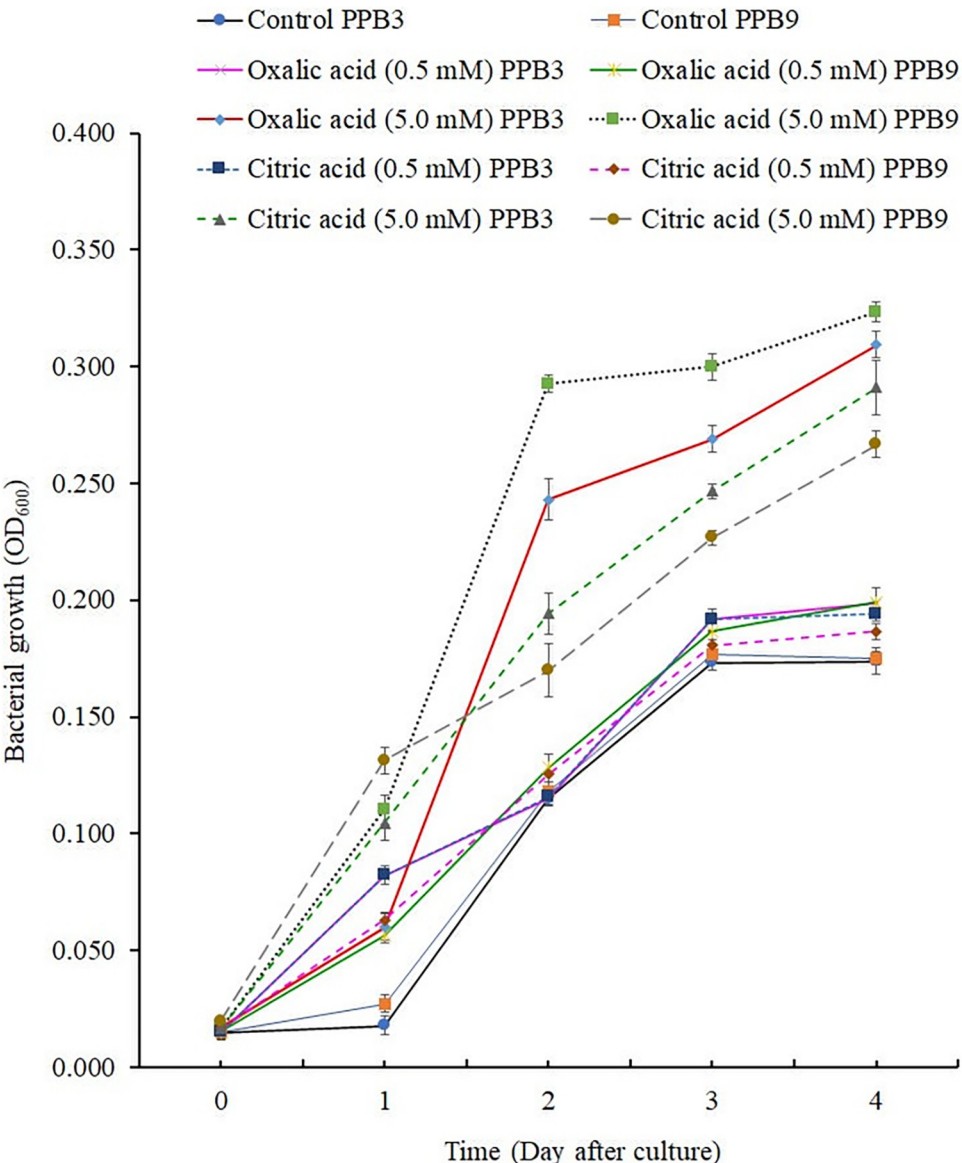

**Fig 1. Growth of *Stenotrophomonas maltophilia* PPB3 (PPB3) and *Bacillus subtilis* PPB9 (PPB9) on water-yeast broths (WYBs), supplemented with either oxalic acid or citric acid as a carbon source at concentrations of 0.5 mM and 5.0 mM.** Borosilicate glass test tubes containing 20 ml WYB supplemented with oxalic acid at concentrations of 0.5 mM [Oxalic acid (0.5 mM) PPB3; Oxalic acid (0.5 mM) PPB9], and 5.0 mM [Oxalic acid (5.0 mM) PPB3; Oxalic acid (5.0 mM) PPB9] were prepared in triplicate for each culture. Another set of borosilicate glass test tubes containing 20 ml WYB supplemented with citric acid at concentrations of 0.5 mM [Citric acid (0.5 mM) PPB3; Citric acid (0.5 mM) PPB9], and 5.0 mM [Citric acid (5.0 mM) PPB3; Citric acid (5.0 mM) PPB9] were prepared in triplicate for each culture. In addition, WYBs without organic acid addition were included as controls (Control PPB3 and Control PPB9). The broths were inoculated with pre-grown bacterial inocula and incubated on a shaker at 120 rpm at 28°C for four days. The optical density of the cultures was measured daily at 0-, 1-, 2-, 3-, and 4-days post-inoculation. Each point in the line graph represents the mean value of three replicates ($n = 3$). Vertical lines denote the standard errors.

at each time point, starting from 2 to 4-day post-inoculation. Between the two carbon sources at 5.0 mM concentration, PPB3 and PPB9 showed significantly higher growth in broth supplemented with oxalic acid than citric acid. In oxalic acid at a 5.0 mM concentration, *B. subtilis* PPB9 showed significantly higher cell densities than *S. maltophilia* PPB3 on day 2 and 3, while

**Table 2. Plant growth-promoting (PGP) and biocontrol traits of elite bacterial isolates.**

| PGP and biocontrol trait | Bacterial Isolates | |
|---|---|---|
| | *Stenotrophomonas maltophilia* PPB3 | *Bacillus subtilis* PPB9 |
| Phosphate solubilization | +++ | ++ |
| Production of indole 3-acetic acid (μg/ml) | 18.47±0.27 | 36.24±0.63 |
| Nitrogen fixation | ++ | ++ |
| Siderophore | ++ | ++ |
| Chitinase | ++ | + |
| Hydrogen cyanide (HCN) | +++ | ++ |
| 1-aminocyclopropane-1-carboxylate (ACC) deaminase | ++ | +++ |
| Biofilm (OD$_{600}$) | 0.18±0.01 | 0.26±0.02 |
| *Antagonism in dual culture against (% inhibition of mycelial growth) | | |
| *Rhizoctonia solani* | 76.04±0.67 | 89.52±0.69 |
| *Phytophthora capsici* | 88.04±0.36 | 77.13±0.42 |
| *Sclerotinia sclerotiorum* | 82.81±0.58 | 70.76±0.11 |

+, positive; −, negative result for the test. For phosphate solubilization, siderophore production and chitinase production: +, zone of clearance <0.2 mm; ++, zone of clearance 0.2–0.4 mm; +++, >0.4 mm. For N$_2$ fixation: +, growth of the bacterium in the N$_2$-free medium; −, no growth of the bacterium in the N$_2$-free medium. For ACC deaminase production: +++ good; ++ medium; + slight.

*The antagonism against fungal pathogens was measured as percent inhibition of radial growth of the fungal pathogens by PGPR antagonists in a dual plate assay. Values are means ± SE (*n* = 3).

statistically similar cell densities were recorded for the isolates on day 4 (S1 Table in S1 File). These results indicated that oxalic acid at a higher concentration (5.0 mM) supported the growth of both bacteria as a carbon source, which was significantly superior to citric acid treatment at a concentration of 5.0 mM in the WYB.

### *In vitro* plant-growth-promoting and biocontrol traits of elite antagonists

The results of Table 2 revealed that multiple plant-growth-promoting and biocontrol activities were displayed by both *S. maltophilia* PPB3 and *B. subtilis* PPB9. Phosphate solubilization was noted with both isolates, while *S. maltophilia* PPB3 exhibited the highest phosphate solubilization. The IAA production by *S. maltophilia* PPB3 and *B. subtilis* PPB9 ranged from 18.47 to 36.24 μg/ml. Both isolates showed growth in an N$_2$-free medium. Likewise, siderophores were produced by both isolates. Production of HCN and chitinase was detected in PPB3 and PPB9. Both *S. maltophilia* PPB3 and *B. subtilis* PPB9 used ACC as a nitrogen source, implying that these isolates were ACC deaminase positive. Both isolates produced biofilm on glass tube surfaces, although variably. Compared to *B. subtilis* PPB9, a higher adherence was observed by the isolate *S. maltophilia* PPB3. Both isolates were antagonistic and inhibitory to multiple plant pathogenic fungi; *R. solani*, *P. capsici*, and *S. sclerotiorum* (Table 2).

### *In vitro* effect of rhizobacteria on seed germination and seedling vigour of tomato

The *in vitro* tests showed that the seed treatments with PGPR strains had improved seed germination (S2 Table in S1 File) and seedling vigour than the control [Fig 2A]. The germination percent of tomato seeds treated with *S. maltophilia* PPB3 and *B. subtilis* PPB9 was enhanced by 15.11% and 20.27%, respectively, over the value recorded with control (S2 Table in S1 File). Moreover, the germination appeared faster in PGPR-treated seeds than in control seeds (data not shown). The seedling length was significantly higher in seedlings from PPB3 and

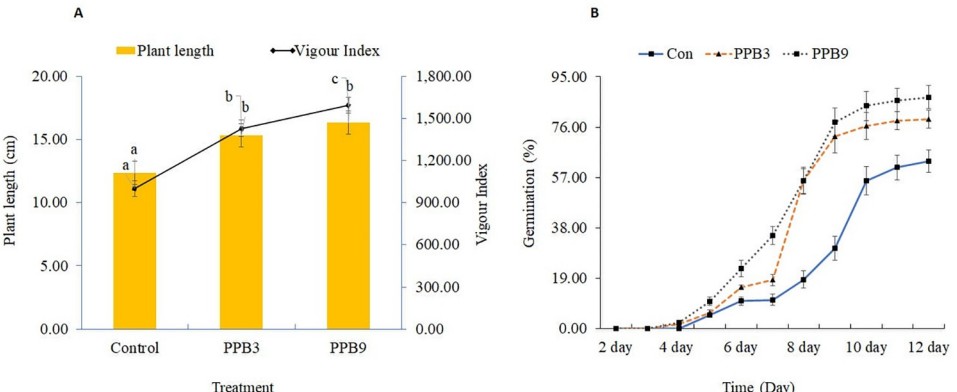

**Fig 2. Effect of rhizobacteria on germination, seedling length and seedling vigour of tomato.** Bacterial seed treatment was done by soaking seeds in the cell suspension of *Stenotrophomonas maltophilia* PPB3 (PPB3) and *Bacillus subtilis* PPB9 (PPB9). Seeds treated with sterile distilled water served as the non-bacterized control (Control). Bacterized and non-bacterized tomato seeds were placed on moistened filter paper in 9-cm Petri dishes. After seven days, the germination percentage, seedling length and vigour (vigour index = % germination × total plant length) were recorded (A). Bacterized and non-bacterized tomato seeds were evaluated for germination in a pot experiment. Fifteen pots (11.50 cm × 15.0 cm) filled with sterilized soils were prepared for each treatment, and each was sown with five seeds. Pots were placed in a growth room at 24˚C temperature and a 16/8 h photoperiod. Germination was recorded daily for 12 days (B). Values are means ± standard errors. Fisher LSD test ($P < 0.05$) was done to indicate significant differences among treatments.

PPB9-treated seeds [Fig 2A], resulting in a 23.99% and 32.16% increase, respectively, compared to control (S2 Table in S1 File). Seed bacterization with strain PPB3 and PPB9 also significantly improved the vigour index of tomato seedlings. The vigour index of tomato seedlings ranged from 1590 to 1000, where the highest seedling vigour index was recorded in seed bacterization by *B. subtilis* PPB9, and the lowest was in control. These findings indicate that seed biopriming with *S. maltophilia* PPB3 and *B. subtilis* PPB9 is potentially effective for promoting germination and vigour in tomatoes. Germination and vigour improvement is considered a vital selection trait for biological inoculants, as it offers an early advantage to the plant that could be vital for its subsequent development.

### *In vivo* effect of rhizobacteria on germination and growth of tomato

Significant improvement in germination was observed in pot experiments with both rhizobacteria treatments in tomatoes compared to non-inoculated controls. Seeds treated with rhizobacteria resulted in quicker and higher germination than non-treated seeds. Tomato seeds sown in control pots started germination five days after sowing, while 1–2% of PPB3 and PPB9-treated seeds germinated four days after sowing. On average, 63% of tomato seeds germinated in the control treatment 12 days after sowing [Fig 2B]. Both bacterial treatments showed similar germination curves, significantly promoting higher germination than the control. Inoculation with strain PPB3 and PPB9 enhanced germination by 25% and 38%, respectively, over the control (S3 Table in S1 File).

Seed bacterization with PPB3 and PPB9 also positively affected plant growth, resulting in significant differences in root and shoot growth in the inoculated plants relative to the control. Treatment with PPB3 and PPB9 increased shoot length by 23% and 29%, shoot fresh biomass by 42% and 77%, shoot dry biomass by 61% and 76%, root length by 25% and 48%, fresh root biomass by 53% and 73%, dry root biomass by 43% and 68%, respectively (Table 3). Compared to control, the leaf number was significantly higher in plants inoculated with strains PPB3 (19%) and PPB9 (32%). The SLA increased by 35% and 41% due to treatment with PPB3 and

**Table 3. Effect of *Stenotrophomonas maltophilia* PPB3 and *Bacillus subtilis* PPB9 treatment on the growth of tomato plants in pots.**

| Treatment | Shoot | | | Root | | | Leaf | | |
|---|---|---|---|---|---|---|---|---|---|
| | Length (cm)* | Fresh weight (g) | Dry weight (g) | Length (cm) | Fresh weight (g) | Dry weight (g) | Number | Specific Leaf Area (m² kg⁻¹) | Total chlorophyll (mg g⁻¹ FW) |
| Control | 25.34±0.65a** | 8.76±0.51a | 0.81±0.02a | 15.34 ±0.47a | 1.26±0.03a | 0.16±0.01a | 4.85 ±0.12a | 22.95±1.26a | 20.23±0.53a |
| *Stenotrophomonas maltophilia* PPB3 | 31.31±1.35b (23.56%)*** | 12.45 ±0.61b (42.12%) | 1.31±0.02b (61.73%) | 19.20 ±0.56b (25.16%) | 1.93±0.02b (53.17%) | 0.23±0.01b (43.75%) | 5.81 ±0.07b (21.03%) | 32.16±2.07b (35.77%) | 30.18±1.10b (49.18%) |
| *Bacillus subtilis* PPB9 | 32.77±0.90c (29.32%) | 15.56±0.58c (77.63%) | 1.43±0.04c (76.54%) | 22.71 ±0.61c (48.04%) | 2.19±0.04c (73.81%) | 0.27±0.01c (68.75%) | 6.42 ±0.08c (32.37%) | 32.45±2.68b (41.39%) | 31.03±0.96 (53.39%) |

*The plant parameters were measured 6 weeks after growing in the pots. Values represent mean ± SE ($n = 3$); one replication consists of five plants.

**Different letters denote significant differences among treatments for each row according to Fisher's LSD test ($P < 5\%$).

*** Data in parenthesis indicate the increase over control.

PPB9, respectively. Total leaf chlorophyll content significantly increased by 49% and 53% in plants from seeds treated with PPB3 and PPB9, respectively (Table 3).

### Effect of rhizobacteria on plant nutrient elements

Significant effect of PGPR treatment in comparison to control was only observed for N, P, and K concentrations. Seed bacterization with PPB3 significantly enhanced N, P, and K concentrations by 48%, 36%, and 29%, respectively, over the control (Table 4). Similarly, N, P, and K concentrations were significantly increased by 73%, 64%, and 68%, respectively, in PPB9-treated plants compared to the control plants. The concentrations of Ca2+, Mg2+, Na+, and Fe did not vary significantly between PGPR treatments and control.

### Biocontrol efficacy of PGPR against *Sclerotium rolfsii*

**Seedling assays.** The results in Fig 3. show that the development of damping-off was fastest in control (unprotected) seedling populations of tomato inoculated with *S. rolfsii*. At the end of the assays, about 74% of the control seedlings had been infected by damping-off [Fig 3A]. Seed bacterization with *S. maltophilia* PPB3 and *B. subtilis* PPB9 resulted in significant pathogen control, causing a delay in symptom development accompanied by a lower number of seedlings with damping-off symptoms. Damping-off symptoms were observed from seeding

**Table 4. Effects of *S. maltophilia* PPB3 and *B. subtilis* PPB9 strains on nutrient concentrations in tomato leaves.**

| Treatment category | Nutrient elements* | | | | | | |
|---|---|---|---|---|---|---|---|
| | %N | P | K | Ca2+ | Mg2+ | Na | Fe |
| Control | 1.68±0.11c** | 2.22±0.18c | 22.16±1.98c | 10.56±1.10ab | 5.08±0.31ab | 0.65±0.05ab | 0.06±0.01a |
| *Stenotrophomonas maltophilia* PPB3 | 2.48±0.13b (47.62%)*** | 3.01±0.26ab (35.59%) | 28.65±2.21b (29.29%) | 11.24±1.16ab (11.73%) | 5.12±0.48ab (0.79%) | 0.68±0.05ab (4.62%) | 0.07±0.01a (16.67%) |
| *Bacillus subtilis* PPB9 | 2.91±0.16a (73.21%) | 3.64±0.28a (63.96%) | 37.31±2.24a (68.37%) | 11.45±1.19a (13.82%) | 5.91±0.56a (16.34%) | 0.72±0.06a (10.77%) | 0.06±0.01a (0.00%) |

* Nutrients were analyzed from oven-dried shoots of plants grown in pots for six weeks. Data were expressed in mg g⁻¹ of dry weight or % for N.

**Values are mean ± standard errors, $n = 3$. Each replication consisted of 5 samples. Different letters denote significant differences among treatments for each column according to Fisher's LSD test ($P < 5\%$).

** Data in parenthesis indicate the increase over control.

until the fourth-week post-sowing. At the end of the experiment, there were 77.15% and 64.93% reductions in damping-off infection by *S. maltophilia* PPB3 and *B. subtilis* PPB9, respectively, over unprotected controls (S4 Table in S1 File). Additionally, a parallel decrease in the number of *S. rolfsii* sclerotia formed on the surfaces of the soils was observed in bacteria-protected seed trays (PPB3; 2.74/cm$^2$ and PPB9; 4.97/cm$^2$) compared to unprotected controls (15.26/cm$^2$) [Fig 3B]. The disease and sclerotia also developed at the lowest rate in inoculated plants treated with Provax 200. No symptom of damping-off developed in the seedlings grown from Provax 200 treatments until the sixth-week post-sowing. At the seventh week post-sowing, only 4.45% of seedlings from Provax 200 treatments showed damping-off symptoms. The lowest number of sclerotia (0.85/cm$^2$) was formed on soil surfaces in Provax 200-treated seed trays (S5 Table in S1 File).

**Potted assays.** Control tomatoes (unprotected) inoculated with *S. rolfsii* showed the first development of southern blight disease, which appeared at the third week (5-week age) post-transplanting [Fig 4A] and increased progressively with time. Most of the plants were severely affected by *S. rolfsii*, and the disease index was recorded to be 91% at the tenth week (12-week age) post-transplanting. Moreover, a significant and steady increase in the numbers of sclerotia was observed in unprotected inoculated plants, rising from about 1 per cm$^2$ to 21 per cm$^2$ soil surface within seven weeks [Fig 4B]. In contrast, introducing PPB3 and PPB9 into the root system effectively controls the disease. The average disease index of PPB3 and PPB9-treated plants with fungal infection was 25% and 40%, respectively. The percent protection achieved by PPB3 and PPB9 treatment against the disease over control was 72% and 56%, respectively (S6 Table in S1 File). Moreover, there was a parallel decrease in the number of *S. rolfsii* sclerotia formed on the surfaces of the soils in bacteria-protected plants (PPB9; 2.12/cm$^2$ and PPB9; 5.60/cm$^2$ soil surface [Fig 4B]. The inoculated plants treated with Provax 200 showed no symptoms of disease and suffered no mortality. The fungicide-protected plants remained healthy throughout the experiment. No formation of sclerotia was observed on the surface of the soil drenched with Provax 200 (S7 Table in S1 File).

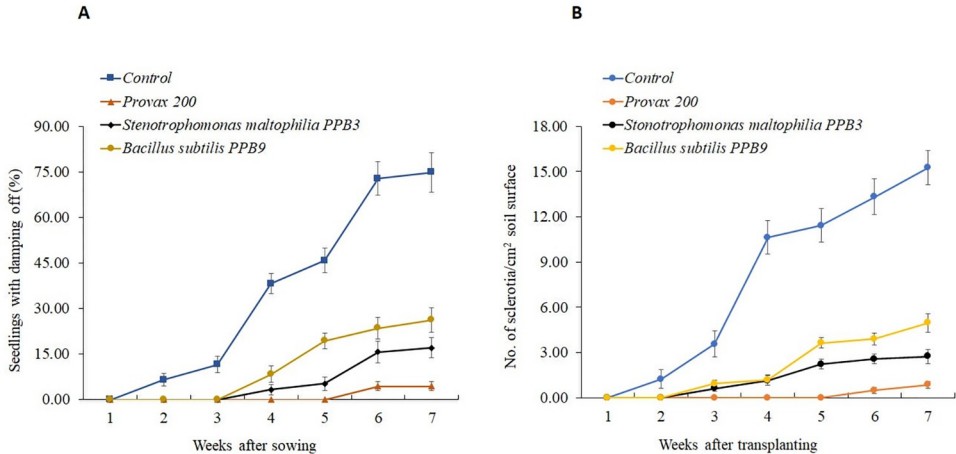

**Fig 3. Development of damping-off and sclerotia caused by *Sclerotium rolfsii* in protected and unprotected seedlings of tomato in seed trays.** Bacteria-protected tomato seedlings against the pathogen were obtained by treating seeds with S*tenotrophomonas maltophilia* PPB3 and *Bacillus subtilis* PPB9. Seedlings from Provax 200 (0.3% w/v)-treated seeds were considered fungicide-protected checks, while those from sterilized-treated water seeds were deemed to be unprotected controls (Control). After autoclaving, field soil mixed with *S. rolfsii* inocula was placed in seed trays (20×10×5 cm). Each tray was sown with 30 tomato seeds and grown in a growth room at 24˚C temperature and a 16/8 h photoperiod for seven weeks. Seedlings with damping-off symptoms were counted weekly (A). The number of sclerotia formed on soil surfaces was counted and expressed per cm$^2$ surface area (B). In each graph, each point in the line represents the mean value within each treatment category. Vertical lines denote the standard errors.

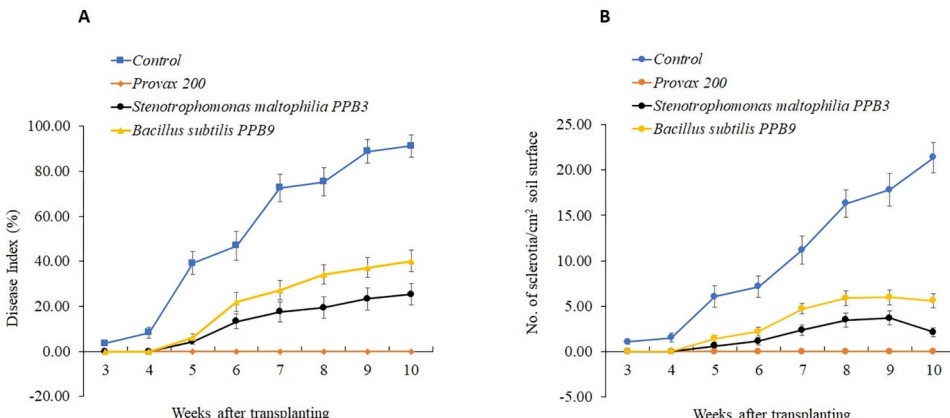

**Fig 4. Development of southern blight disease and sclerotia caused by *Sclerotium rolfsii* in protected and unprotected tomatoes in pots.** Seedlings were protected by rhizobacteria by dipping the seedling roots in the cell suspension of *Stenotrophomonas maltophilia* PPB3 and *Bacillus subtilis* PPB9 before transplanting in soil containing *Sclerotia rolfsii* inoculum. Seedlings of unprotected control were dipped in the sterilized water (Control). Soils drenched with Provax 200 (0.3% w/v) at transplanting and one-week post transplanting were considered fungicide-protected checks. Disease developments on each plant were rated every week from 2 weeks after transplanting, where 0 = no symptoms, 1 = <25% of leaves with symptoms, 2 = 26 to 50% of leaves with symptoms, 3 = 51 to 75% of leaves with symptoms, 4 = 76 to 100% of leaves with symptoms and 5 = plant dead. The disease index was calculated (A). An abundance of sclerotia of *S. rolfsii* in the soils of protected and unprotected adult potted plants was also calculated (B). In each graph, each point in the line represents the mean value within each treatment category. Vertical lines denote the standard errors.

**Field assays.** The results of field-grown tomatoes showed that Southern blight disease appeared faster in plants grown in inoculated control plots than in those grown in other plots. The disease symptoms in the unprotected control plants were seen within two weeks of inoculation and increased progressively with time. The disease index in these plants was estimated to be about 57.05 at 12-week post-inoculation. On the other hand, inoculated plants were protected significantly with *S. maltophilia* PPB3 and *B. subtilis* PPB9, showing the first symptoms of southern blight within four weeks of inoculation. The protected plants remained almost healthy throughout the experiment and suffered less mortality, recording a disease index of 17.34 and 23.83 in plants treated with *S. maltophilia* PPB3 and *B. subtilis* PPB9, respectively, at 12-week post-inoculation. The percent protection achieved by PPB3 and PPB9 against the disease over inoculated controls was about 69% and 58%, respectively. In addition, these plants were significantly taller (PPB3; 47.58 cm and PPB9; 44.09 cm) and set higher fruit yield (PPB3; 27.87 t/ha and PPB9; 20.81 t/ha) than the pathogen inoculated control (Height- 31.26 cm; Yield- 11.53 t/ha) (Table 5). On the other hand, inoculated plants protected with Provax 200 showed the least disease index (9.78) of southern blight throughout the experiment and suffered no mortality. These fungicide-protected plants were also significantly taller and gave higher fruit yields (Table 5) than the inoculated control plants. This shows that *S. maltophilia* PPB3 and *B. subtilis* PPB9 are potential biocontrol agents that can protect against southern blight disease and improve the yield of tomato plants.

## Tomato root colonization by PGPR strains

The root colonization assays showed that both rhizobacteria successfully colonized the roots of tomato plants. The total root population densities of *B. subtilis* PPB9 and *S. maltophilia* PPB3 were $47\times 10^7$ and $25\times 10^7$ CFU/g root fresh weight, respectively, during the first week of seedling growth [Fig 5]. However, the total root population densities of B. *subtilis* PPB9 and *S.*

**Table 5. Effect of rhizobacterial treatments on disease severity and yield-contributing parameters in *Sclerotium rolfsii*-inoculated tomato cv. Minto Super F1 in the field.**

| Treatment category | Disease index* | Plant height (cm)** | Yield (t/ha)*** |
|---|---|---|---|
| **Uninoculated negative control** | 12.34±0.41b**** | 41.08±2.14b | 21.64±1.37b |
| **Pathogen inoculated control (*Sclerotium rolfsii*)** | 57.05±3.17e | 31.26±1.19a | 11.53±0.63a |
| ***Stenotrophomonas maltophilia* PPB3+ *S. rolfsii*** | 17.34±1.51c (-69.61%)***** | 47.58±3.08c (52.21%) | 27.87±1.45c (141.72%) |
| ***Bacillus subtilis* PPB9+ *S. rolfsii*** | 23.83±1.11d (-58.23%) | 44.09±2.46b (41.04%) | 20.81±1.15b (80.49%) |
| **Provax 200+ *S. rolfsii*** | 9.78±0.33a (-82.86%) | 42.89±2.63b (37.20%) | 22.92±1.89b (107.46%) |

*Disease developments were rated at 12 weeks after transplanting using 0–5 scale (0 = no symptoms, 1 = <25% of leaves with symptoms, 2 = 26 to 50% of leaves with symptoms, 3 = 51 to 75% of leaves with symptoms, 4 = 76 to 100% of leaves with symptoms and 5 = plant dead) and disease index was calculated using the formula [24]. Values are means ± standard errors (*n = 3*).

**Plant height was recorded at harvest.

***Fruits were harvested at several spells, and yield was calculated.

****Different letters denote significant differences among treatments for each column according to Fisher's LSD test (*P < 5%*).

***** Data in parenthesis indicate the increase or decrease over pathogen inoculated control.

*maltophilia* PPB3 reached 240× $10^7$ and 165× $10^7$ CFU/g roots fresh weight, respectively, after seven weeks [Fig 5]. For both bacteria, root population densities were significantly higher in the bottom part than in the middle and upper part. The population differences among the three root segments increased with plant age and total bacterial population. At seven weeks, the root population of PPB9 in the bottom root parts was two and four folds higher than in the middle and upper root parts, respectively (S8 Table in S1 File). Similarly, the bottom root parts had two and three folds higher PPB3 populations at the same growth stage than the middle and upper root parts, respectively.

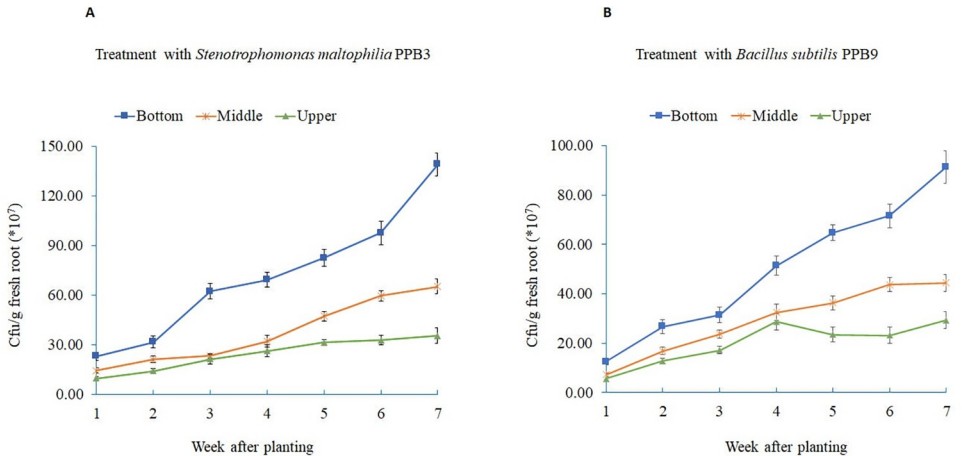

**Fig 5. Population of *Stenotrophomonas maltophilia* PPB3 and *Bacillus subtilis* PPB9 in the upper (toward stem) (top), middle, and lower (toward root tip) (bottom) segments of 1-, 2-, 3-, 4-, 5-, 6- and 7-week-old tomato seedlings.** Seeds were treated with the bacteria and were grown in pots for 12 weeks. Roots were harvested at 1, 2, 4, 5, 6, and 7 weeks after sowing and cut into the top, middle, and bottom regions. Root tissues of each segment were separately homogenized in sterilized distilled water. Appropriate dilutions of the homogenates were plated onto Yeast Peptone Agar (YPA) media. The plates were incubated overnight at 28°C, and the number of colony-forming units (CFU) per gram of root tissue was determined. Data are presented as CFU g$^{-1}$ fresh weight, each from three sets of five roots harvested from three plants at each time point.

## Discussion

Exploiting rhizobacterial antagonists showing beneficial influence on the growth, development, and protection of plants against diseases facilitates sustainable approaches in agriculture [23–25]. Six bacterial strains were screened in various dual culture assays, from which *S. maltophilia* PPB3 and *B. subtilis* PPB9 were selected for superior antagonistic activity against *S. rolfsii*. In many studies, *B. subtilis* shows a substantial antagonistic effect against many common fungal pathogens, including *S. rolfsii* [26]. However, strains of *S. maltophilia* have recently been reported as antagonists of several important plant pathogens [12,27], but not against *S. rolfsii*. These two promising strains were tested further under *in vitro*, greenhouse, and field conditions for their plant-growth-promoting and biocontrol potentials. Tomato seed bacterization with these bacteria promoted rapid and even seed germination and improved seedling vigour. The introduction of bacteria into the root systems effectively enhanced the plant growth and leaf photosynthetic pigments of tomatoes. The bacteria in this study also appeared to have significant value as efficient biocontrol agents to protect seedlings, potted plants, and field-grown tomatoes against *S. rotfsii*. Both *S. maltophilia* PPB3 and *B. subtilis* PPB9 contributed significantly to the plant-growth promotion, fruit formation, and yield improvement in tomatoes. Bacteria within the genus *Bacillus* have often been illustrated as effective PGPR, and biological control agents of plant diseases in various crop species, including tomato [12,28–33], but references to the genus *Stenotrophomonas* are less common in tomato. These results confirm that *Stenotrophomonas* can be a potential PGPR and biological control agent of *S. rotfsii* in tomatoes. However, judicious use of the bacterium is required as the species is also reported as an opportunistic human pathogen [34].

Improvement of germination, seedling vigour, plant growth, and yield by *S. maltophilia* PPB3 and *B. subtilis* PPB9 in this study can be attributed to the enhanced IAA production, ACC-deaminase activity, photosynthesis, and nutrient mobilization. IAA, the major auxin, is a crucial phytohormone produced by many PGPR. Seed treatment with such IAA-producing rhizobacteria promotes germination and plant growth [12,35]. The rhizobacterial isolates listed in the present study were capable of producing IAA, ranging from 18.47 to 36.24 μg/ml culture medium. IAA is shown to be involved in the early stages of seed germination [36]. It is also found that exogenous IAA treatment increases germination rate, seedling length, and seedling weight in various plant species [37,38]. Seed vigour and germination are directly related to membrane system integrity, particularly mitochondrial repair [39]. IAA is suggested to repair the membrane system integrity and improve the properties of seed vigour and germination [40]. IAA is also known to affect plant growth by promoting root development and apical dominance [41].

*In vitro* ACC-deaminase activity was shown by the rhizobacterial isolates used in the present study. ACC-deaminase is a crucial enzyme found in some PGPR. Being an immediate biosynthetic precursor of ethylene, ACC directly regulates ethylene production in higher plants [42]. The ACC-deaminase enzyme present in some PGPR breaks down ACC into ammonia and α-ketobutyrate, reducing the substrate for ethylene production and stimulating plant growth by minimizing the ethylene-induced plant growth inhibition [43]. Plant growth-promoting *B. subtilis* and *S. maltophilia* with ACC deaminase activity was previously reported [44,45]. The ACC-deaminase-producing PGPR have also been shown to significantly improve the plant photosynthetic apparatus (leaf chlorophyll). The higher photosynthetic potential may lead to higher carbon assimilation and stimulated growth in plants [24].

The tested bacteria showed *in vitro* phosphate solubilization and $N_2$ fixation ability. The two traits have been known as the potential plant growth-promoting mechanisms by many PGPR including, strains of *B. subtilis* and *S. maltophilia* [12,46,47]. This is supported by the

fact that the PGPR inoculation improved concentrations of plant nutrient elements such as N, P, and K in the present study. Some of the earlier studies with *S. maltophilia* and *B. subtilis* have reported higher N, P, and K contents in the inoculated plants confirming our data in the present work [47,48]. The increase in N, P, and K concentration in tomato leaves could additionally be explained by one or more indirect PGPR actions such as (1) increase in root length and root surface area for nutrient uptake, (2) activation of different metabolic processes that mobilize nutrients, (3) sorption equilibrium shift and increase in net nutrient transfer into the soil solution, or (4) microbial biomass output in the rhizosphere [49]. On the other hand, rhizobacterial treatment did not significantly improve the $Ca^{2+}$, $Mg^{2+}$, Na, and Fe concentrations in the inoculated plants compared to the control. Similar tendencies were observed in an earlier study with different PGPR inoculation in tomato plants [50], demonstrating that these PGPR are specifically beneficial for improving plant N, P, and K nutrition. Soil nutrient availability and plant nutrient uptake are usually influenced by a range of interrelated biotic and abiotic factors. The complex interactions among these factors might result in higher positive effects of the PGPR on improving plant N, P, and K concentrations than $Ca^{2+}$, $Mg^{2+}$, Na, and Fe.

Application of *S. maltophilia* PPB3 and *B. subtilis* PPB9 significantly controlled the disease development on tomatoes caused by *S. rolfsii*. Both rhizobacteria showed the ability to inhibit the mycelial growth of several plant pathogens, including *S. rolfsii*. Moreover, the culture filtrates of the bacteria inhibited the mycelial growth of *S. rolfsii* with the increase of the concentrations. These results suggest that the antifungal compounds produced by the bacteria were responsible for the suppression of plant pathogen and the disease. Antibiosis by bacterial antagonists is implicated as a typical mechanism in their biocontrol activity against *S. rolfsii* [51]. Additionally, other data of this study demonstrated that the tested rhizobacteria inhibited oxalic acid production of *S. rolfsii* and showed abilities to use oxalic acid as a growth substrate. Oxalic acid is one of the critical determinants of virulence and pathogenicity of *S. rolfsii* [52]. Neutralizing this pathogenicity factor of *S. rolfsii* by antagonists may appear to be significant for the control of the disease. In earlier studies, oxalate degrading or oxalotrophic soil bacteria were shown to provide significant protection against oxalate-producing pathogens *S. rolfsii*, *S. sclerotiorum*, and *Botrytis cinerea* [53,54]. Therefore, oxalotrophic isolates of *S. maltophilia* and *B. subtilis* are of significant value in protecting tomato plants against *S. rolfsii*. Equally, the production of siderophores and cell wall degrading enzymes by the bacteria was partially or solely responsible for suppressing *S. rolfsii*. Siderophores producing PGPR are recognized as resilient antagonists against phytopathogenic fungi [55]. Bacterial siderophores deprive plant pathogens of iron under iron-limiting conditions, promoting plant growth through pathogen control [56]. Similarly, the volatile antibiotic HCN is fungitoxic and accounted for substantial biocontrol of phytopathogenic fungi by cyanogenic bacteria [57]. Chitinolytic enzymes such as chitinases that are produced by antagonists break down chitin, a principal constituent of hyphae of pathogenic fungi. In several studies, chitinolytic bacteria showed *in vitro* inhibitory effect on mycelial growth of *S. rolfsii*, and plant treatment with these isolates inhibited the disease completely [58,59]. These agree with our results demonstrating that the biocontrol potential of *B. subtilis* and *S. maltophilia* against *S. rolfsii* might be due to the activity of one or more antimicrobial mechanisms.

The rhizobacterial antagonists showed early establishment on the tomato roots as indicated by increases in the abundance of the antagonists from $25–47\times 10^7$ to about $138–240\times 10^7$ colony-forming units per g root tissues within the seven weeks of the experiment. This may have contributed to the rapid colonization of the rhizosphere and the rhizoplane. The maximum density of bacteria appeared higher at the root tip. The root tip or root growth zone is particularly active in nutrient uptake and may provide a favorable niche for profuse microbial growth

and activity [60,61]. These antagonists also showed consistency in forming a biofilm, which enabled them to make sufficient attachment to the root surface and exert plant growth promotion and biocontrol activities. Earlier studies have also established the role of biofilm and consequential root colonization of PGPR in promoting plant growth and controlling the disease [62,63].

## Conclusions

The present study has demonstrated that the rhizobacteria *S. maltophilia* PPB3 and *B. subtilis* PPB9 are superior among the six strains in antagonism against tomato southern blight pathogen *S. rolfsii*. The selected two strains possess multiple plant-growth-promoting and biocontrol traits. Inoculating tomato plants with these bioagents effectively stimulated growth, managed *S. rolfsii*, and enhanced yields. Thus, introducing these PGPR strains in bio-fertilization and biocontrol could have a positive impact on tomato cultivation.

## Supporting information

**S1 File.**
(DOCX)

## Acknowledgments

Financial supports from Ministry of Science and Technology, Bangladesh are greatly acknowledged. We also sincerely wish to thank students and staff members in the College of Agricultural Sciences, IUBAT and Department of Plant Pathology, BSMRAU for their support and assistance in the research works.

## Author Contributions

**Conceptualization:** M. Motaher Hossain.

**Data curation:** Farjana Sultana.

**Formal analysis:** Farjana Sultana.

**Funding acquisition:** M. Motaher Hossain.

**Investigation:** Farjana Sultana, M. Motaher Hossain.

**Methodology:** Farjana Sultana, M. Motaher Hossain.

**Project administration:** M. Motaher Hossain.

**Supervision:** M. Motaher Hossain.

**Writing – original draft:** Farjana Sultana.

**Writing – review & editing:** M. Motaher Hossain.

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
