## [Decision Letter · Decision Letter 0]

10 Jun 2021

PONE-D-21-13656

Assessing the potentials of bacterial antagonists for plant growth promotion, nutrient acquisition, and biological control of Southern blight disease in tomato

PLOS ONE

Dear Dr. Hossain,

Thank you for submitting your manuscript to PLOS ONE. After careful consideration, we feel that it has merit but does not fully meet PLOS ONE’s publication criteria as it currently stands. Therefore, we invite you to submit a revised version of the manuscript that addresses the points raised during the review process.

We look forward to receiving your revised manuscript.

Kind regards,

Ying Ma, Ph.D.

Academic Editor

PLOS ONE

Journal Requirements:

2. Please include a copy of Table 1 which you refer to in your text on page 15.  (Please note that two tables in the manuscript have the title of 'table 4'.)

Reviewers' comments:

Reviewer's Responses to Questions

**Comments to the Author**

1. Is the manuscript technically sound, and do the data support the conclusions?

Reviewer #1: Yes

Reviewer #2: Partly

2. Has the statistical analysis been performed appropriately and rigorously? 

Reviewer #1: Yes

Reviewer #2: No

3. Have the authors made all data underlying the findings in their manuscript fully available?

Reviewer #1: Yes

Reviewer #2: No

4. Is the manuscript presented in an intelligible fashion and written in standard English?

Reviewer #1: Yes

Reviewer #2: No

5. Review Comments to the Author

Reviewer #1: I have read the article titled “Assessing the potentials of bacterial antagonists for plant growth promotion, nutrient acquisition, and biological control of Southern blight disease in tomato” by Sultana and Hossain.

This article analyses the potential of different rhizobacteria to promote plant growth and control plant disease caused by S. rolfsii in tomato. This is a very detailed study of different bacterial traits involved in plant growth and biocontrol. Moreover, authors explore the performance of the isolates through several experimental conditions from growth chambers to field conditions and this is extremely important because the use of microorganisms usually varies a lot with these conditions. In this way, authors were able to identify isolates with good performances in every condition with potential to develop bioinoculants for tomato.

I think this work could be publish in Plos One, but there are a few comments and questions, which I mention below, that authors should address in order to accept this manuscript.

Section 2.4.4

Gram-positive bacteria do not growth in classic CAS plates, you tested several Bacillus strains, how did you manage to perform this experiment?

Section 2.4.9

In this section you aim to determine biofilm formation by bacterial isolates. However, the methodology you mentioned is used for pellicle (one type of biofilm) determination, why did you choose this? I think it would be interesting to measure general biofilm formation using crystal violet as described by the same reference authors cited (Haque et al. 2012).

Section 2.5

Please specify the concentration (CFU/OD) of the bacterial preparation that you used to inoculate seeds.

Section 2.8

Plants were watered just with water or did you use nutrient solution?

Section 3.3

The decrease in the production of oxalic acid y the co-cultures of rhizbacteria and S. rolfsii could be due to growth inhibition of the pathogen or to the comsuption of oxalic acid by rhizobacteria. I think it would be interesting to investigate this. For example, you could express the oxalic acid concentration in relation to fungal biomass. Moreover you could check if rhizobacteria are able to growth using oxalic acid as unique carbon source.

Section 3.7

I would remove the first sentence in this section because it is confuse since most of the elements did not change due to inoculation (Fe, Ca, Mg and Na) as authors mentioned in the last sentence of this section.

Page 9 line 22: remove at 28°C

Page 15 line 21: Please include the inhibition % of PPB3

Page 10 line 08: vortexing

Reviewer #2: The author did not summarize the main research and key findings well. Example, they did not state that five strains were screened from which two strains were selected for superior performance. Analysis of the effects of S. maltophilia PPB3 and B. subtilis PPB9 strains on nutrient concentrations in tomato included N, P, K, Ca2+, Mg2+, Na and Fe, but significant differences were only in NPK . The authors identified other literature but did not explain how this study related to previously published research or the justification of all the studies presented. Why test the filtrates? What is the significance of 20%, 50% etc. filtrate concentration? the discussion did not address the significance of the findings

Figure and table captions do not stand alone (independent) but are linked to materials and methods, names are not spelt out on Table or figure captions. example B. subtilis. Qualities of figures need to be improved, legends of figures and tables are not accurate example comparisons within rows instead of columns.

Although the research is technically sound, the presentation need improvement and editing is needed, there are spelling mistakes. It would be difficult for another researcher to reproduce the study with the same methods because the materials do not include enough details, example, what was the environment of the experiment on potted plants? Type of soil, pot size, irrigation, fertilization temperature is important for this disease.

Overall, the writing quality need improvement

6. PLOS authors have the option to publish the peer review history of their article (what does this mean?). If published, this will include your full peer review and any attached files.

Reviewer #1: **Yes: **Fernando M. Romero

Reviewer #2: No

---

## [Author Response · Author response to Decision Letter 0]

27 Aug 2021

Response to Reviewers

First of all, we would like to acknowledge the excellent efforts given by the Editor and Reviewers to analyse and evaluate our manuscript. We carefully checked the comments made by Editor and two Reviewers one by one and incorporated the proposed amendments in our manuscript accordingly. The text of any added subject matter is shown by Track changes Mode in Microsoft word. 

Point-by-point responses to the Editor and Reviewer comments.

Editor’s comments:

We have followed PLOS ONE’s style and formatted the manuscript accordingly.

 2. Please include a copy of Table 1 which you refer to in your text on page 15. (Please note that two tables in the manuscript have the title of 'table 4'.)

We have corrected the Table names.

Reviewer #1:

We would like to thank you for your kind suggestion. We addressed all your recommendations in the revised manuscript. We believe that your suggestion and comments have improved the manuscript significantly. Please see our response to your specific comment regarding this matter.

1. Gram-positive bacteria do not grow in classic CAS plates, you tested several Bacillus strains, how did you manage to perform this experiment?

We have described the experiment elaborately and appropriately in Materials and Methods (Page 10, Line 1-13). 

2. In this section you aim to determine biofilm formation by bacterial isolates. However, the methodology you mentioned is used for pellicle (one type of biofilm) determination, why did you choose this? I think it would be interesting to measure general biofilm formation using crystal violet as described by the same reference authors cited (Haque et al. 2012).

The bacteria of this study produced only air-liquid biofilm (only form pellicle), not solid-air-liquid biofilm. That is why we did not use the Crystal violet method. 

3. Please specify the concentration (CFU/OD) of the bacterial preparation that you used to inoculate seeds.

We specified the concentration of the bacterial treatment in the materials and methods (Page 12, Line 16).

4. Were plants watered using water, or did you use the nutrient solution?

We used only water regularly but use nutrient solution weekly until eight weeks after planting.

5. The decrease in the production of oxalic acid with the co-cultures of rhizobacteria and S. rolfsii could be due to growth inhibition of the pathogen or to the consumption of oxalic acid by rhizobacteria. I think it would be interesting to investigate this. For example, you could express the oxalic acid concentration in relation to fungal biomass. 

We agree with you. That is why according to your suggestion, we have expressed the oxalic acid concentration in relation to fungal biomass (Page 8 Line 1-14). 

6. Moreover, you could check if rhizobacteria are able to growth using oxalic acid as unique carbon source.

We have tested the ability of the two selected rhizobacteria to utilize oxalic acid as a carbon source and added the results in the manuscript. Both rhizobacteria were found to utilize oxalic acid for their growth. (Fig. 1) (page 20 line 12-23)

7. I would remove the first sentence in this section because it is confused since most of the elements did not change due to inoculation (Fe, Ca, Mg and Na) as authors mentioned in the last sentence of this section.

We have removed the first sentence (page 26).

8. Page 11, line 23: remove at 28°C

We have removed the sentence as it was repeated.

9. Page 18, line 16: Please include the inhibition % of PPB3

We have included the % inhibition of PPB3.

10. Page 12, line 13: vortexing

We have corrected it.

Reviewer #2: 

We express sincere gratitude to you for your valuable suggestion. We addressed all your concerns and comments in the revised manuscript. We believe that the quality of the manuscript has been improved substantially by incorporating your suggested revision. Please see our response to your specific comment regarding this matter.

1. The author did not summarize the main research and key findings well. Example, they did not state that five strains were screened from which two strains were selected for superior performance. 

We have summarized the key research findings and included them in the abstract, discussion, and conclusions according to your suggestion.

2. Why test the filtrates? What is the significance of 20%, 50% etc. filtrate concentration? the discussion did not address the significance of the findings.

We have discussed the significance of the findings concerning the antagonistic activity of culture filtrates of bacteria according to your suggestion (Page 33, Line 16-22).

3. Analysis of the effects of S. maltophilia PPB3 and B. subtilis PPB9 strains on nutrient concentrations in tomatoes included N, P, K, Ca2+, Mg2+, Na and Fe, but significant differences were only in NPK. The authors identified other literature but did not explain how this study related to previously published research or the justification of all the studies presented.

We have improved the discussion by citing related articles according to your suggestion (Page 33, Line 1-15).

4. Figure and table captions do not stand alone (independent) but are linked to materials and methods, names are not spelt out on Table or figure captions. example B. subtilis. 

We have improved the legends of all figures and Tables. 

5. Qualities of figures need to be improved, legends of figures and tables are not accurate example comparisons within rows instead of columns.

We have also improved the quality of the figures and amended Table 4 according to your suggestion.

6. Although the research is technically sound, the presentation need improvement and editing is needed, there are spelling mistakes. It would be difficult for another researcher to reproduce the study with the same methods because the materials do not include enough details, example, what was the environment of the experiment on potted plants? Type of soil, pot size, irrigation, fertilization, temperature is important for this disease. Overall, the writing quality need improvement

We have thoroughly revised the manuscript and improved the writing quality of the whole manuscript, including Materials and Methods. In addition, we included necessary information about soil type, pot size, irrigation, fertilization and growth conditions.

---

## [Editor Report · Decision Letter 1]

1 Sep 2021

PONE-D-21-13656R1

Assessing the potentials of bacterial antagonists for plant growth promotion, nutrient acquisition, and biological control of Southern blight disease in tomato

PLOS ONE

Dear Dr. Hossain,

Thank you for submitting your manuscript to PLOS ONE. After careful consideration, we feel that it has merit but does not fully meet PLOS ONE’s publication criteria as it currently stands. Therefore, we invite you to submit a revised version of the manuscript that addresses the points raised during the review process.

We look forward to receiving your revised manuscript.

Kind regards,

Ying Ma, Ph.D.

Academic Editor

PLOS ONE
---

## [Author Response · Author response to Decision Letter 1]

5 Oct 2021

Response to Reviewers

First of all, we would like to admit the outstanding efforts given by the Academic Editor and Reviewers to evaluate our manuscript. We have carefully checked the Reviewers comments and made the necessary amendments. 

Point-by-point responses to the Editor and Reviewer comments

Reviewers' comments:

1. While revising your submission, please upload your figure files to the Preflight Analysis and Conversion Engine (PACE) digital diagnostic tool, https://pacev2.apexcovantage.com/. PACE helps ensure that figures meet PLOS requirements. To use PACE, you must first register as a user. Registration is free. Then, login and navigate to the UPLOAD tab, where you will find detailed instructions on how to use the tool. Please note that Supporting Information files do not need this step.

We have uploaded our figures in PACE, and PACE has confirmed that the figure file meets PLOS requirements. The PACE processed figure files have been included in the resubmission of the manuscript.

---

## [Decision Letter · Decision Letter 2]

8 Mar 2022

PONE-D-21-13656R2Assessing the potentials of bacterial antagonists for plant growth promotion, nutrient acquisition, and biological control of Southern blight disease in tomatoPLOS ONE

Dear Dr. Hossain,

Thank you for submitting your manuscript to PLOS ONE. After careful consideration, we feel that it has merit but does not fully meet PLOS ONE’s publication criteria as it currently stands. Therefore, we invite you to submit a revised version of the manuscript that addresses the points raised during the review process.

We look forward to receiving your revised manuscript.

Kind regards,

Ying Ma, Ph.D.

Academic Editor

PLOS ONE

Journal Requirements:

Reviewers' comments:

Reviewer's Responses to Questions

**Comments to the Author**

1. If the authors have adequately addressed your comments raised in a previous round of review and you feel that this manuscript is now acceptable for publication, you may indicate that here to bypass the “Comments to the Author” section, enter your conflict of interest statement in the “Confidential to Editor” section, and submit your "Accept" recommendation.

Reviewer #1: (No Response)

Reviewer #3: All comments have been addressed

2. Is the manuscript technically sound, and do the data support the conclusions?

Reviewer #1: Yes

Reviewer #3: Partly

3. Has the statistical analysis been performed appropriately and rigorously? 

Reviewer #1: Yes

Reviewer #3: Yes

4. Have the authors made all data underlying the findings in their manuscript fully available?

Reviewer #1: Yes

Reviewer #3: Yes

5. Is the manuscript presented in an intelligible fashion and written in standard English?

Reviewer #1: Yes

Reviewer #3: Yes

6. Review Comments to the Author

Reviewer #1: I have read the revised version of the manuscript titled: “Assessing the potentials of bacterial antagonists for plant growth promotion, nutrient acquisition, and biological control of Southern blight disease in tomato” by Sultana and Hossain.

Authors have improved the manuscript and answered all my previous suggestions, so I think it is ready for publication. However I found two minor issues to address:

Page 20, line 5: Please correct the amount of oxalic acid produced in co-cultive with PBB9, it doesn´t match with the results in Table 1.

Figure 2A: You are trying to show 3 different traits (germination, plant length and vigour index) however you have only 2 vertical axes, so you can´t show the reference for plant length so it is a bit confusing. Since you already show germination in the figure 2B, I would remove this parameter from figure 2A, so it is easier to understand.

Reviewer #3: The MS is now modified as suggested but I suggest to reduce number of references. All the figures and tables have been modified as suggested.

7. PLOS authors have the option to publish the peer review history of their article (what does this mean?). If published, this will include your full peer review and any attached files.

Reviewer #1: **Yes: **Fernando Matias Romero

Reviewer #3: **Yes: **Dr. Arup Kumar Mukherjee

---

## [Author Response · Author response to Decision Letter 2]

10 Mar 2022

Responses to Editor’s and Reviewer’s comments to authors

First of all, we would like to acknowledge the excellent efforts given by the Editor and Reviewers to analyse and evaluate our manuscript. We carefully checked the comments and incorporated the proposed amendments in our manuscript accordingly. The text of any added subject matter is shown by Track changes Mode in Microsoft word. 

Point-by-point responses to the Reviewer comments.

Reviewer #1:

We would like to thank you for your kind suggestion. We have addressed all your recommendations in the revised manuscript. Please see our response to your specific comment regarding this matter.

1. Page 20, line 5: Please correct the amount of oxalic acid produced in co-cultivate with PBB9, it doesn´t match with the results in Table 1.

We have corrected it 

2. Figure 2A: You are trying to show 3 different traits (germination, plant length and vigour index) however you have only 2 vertical axes, so you can´t show the reference for plant length so it is a bit confusing. Since you already show germination in the figure 2B, I would remove this parameter from figure 2A, so it is easier to understand.?

We have amended Figure 2A according to your suggestion. 

Reviewer #2: 

We express sincere gratitude to you for your valuable suggestion. We have addressed your comments in the revised manuscript. Please see our response to your specific comment regarding this matter.

1. The MS is now modified as suggested but I suggest to reduce number of references. 

Following your suggestion, we have reduced the number of references from 79 to 63.

---

## [Decision Letter · Decision Letter 3]

6 Apr 2022

Assessing the potentials of bacterial antagonists for plant growth promotion, nutrient acquisition, and biological control of Southern blight disease in tomato

PONE-D-21-13656R3

Dear Dr. Hossain,

We’re pleased to inform you that your manuscript has been judged scientifically suitable for publication and will be formally accepted for publication once it meets all outstanding technical requirements.

Kind regards,

Ying Ma, Ph.D.

Academic Editor

PLOS ONE

Additional Editor Comments (optional):

Reviewers' comments:

Reviewer's Responses to Questions

**Comments to the Author**

1. If the authors have adequately addressed your comments raised in a previous round of review and you feel that this manuscript is now acceptable for publication, you may indicate that here to bypass the “Comments to the Author” section, enter your conflict of interest statement in the “Confidential to Editor” section, and submit your "Accept" recommendation.

Reviewer #1: All comments have been addressed

Reviewer #3: All comments have been addressed

2. Is the manuscript technically sound, and do the data support the conclusions?

Reviewer #1: (No Response)

Reviewer #3: Yes

3. Has the statistical analysis been performed appropriately and rigorously? 

Reviewer #1: (No Response)

Reviewer #3: Yes

4. Have the authors made all data underlying the findings in their manuscript fully available?

Reviewer #1: (No Response)

Reviewer #3: Yes

5. Is the manuscript presented in an intelligible fashion and written in standard English?

Reviewer #1: (No Response)

Reviewer #3: Yes

6. Review Comments to the Author

Reviewer #1: (No Response)

Reviewer #3: The authors have addressed the queries. The numbers of references have been reduced. The figure ha sbeen corrected. So, it may be considered for acceptance.

7. PLOS authors have the option to publish the peer review history of their article (what does this mean?). If published, this will include your full peer review and any attached files.

Reviewer #1: **Yes: **Fernando M. Romero

Reviewer #3: **Yes: **Dr Arup Kumar Mukherjee

---

## [Editor Report · Acceptance letter]

31 May 2022

PONE-D-21-13656R3 

Assessing the potentials of bacterial antagonists for plant growth promotion, nutrient acquisition, and biological control of Southern blight disease in tomato 

Dear Dr. Hossain:

I'm pleased to inform you that your manuscript has been deemed suitable for publication in PLOS ONE. Congratulations! Your manuscript is now with our production department. 

Kind regards, 

on behalf of

Dr. Ying Ma 

Academic Editor

PLOS ONE